# UNDERSTANDING SHORT-HORIZON BIAS IN STOCHASTIC META-OPTIMIZATION

**Yuhuai Wu**[*]**, Mengye Ren**[*]**, Renjie Liao & Roger B. Grosse**
University of Toronto and Vector Institute
`{ywu, mren, rjliao, rgrosse}@cs.toronto.edu`

## ABSTRACT

Careful tuning of the learning rate, or even schedules thereof, can be crucial to effective neural net training. There has been much recent interest in gradient-based meta-optimization, where one tunes hyperparameters, or even learns an optimizer, in order to minimize the expected loss when the training procedure is unrolled. But because the training procedure must be unrolled thousands of times, the meta-objective must be defined with an orders-of-magnitude shorter time horizon than is typical for neural net training. We show that such short-horizon meta-objectives cause a serious bias towards small step sizes, an effect we term short-horizon bias. We introduce a toy problem, a noisy quadratic cost function, on which we analyze short-horizon bias by deriving and comparing the optimal schedules for short and long time horizons. We then run meta-optimization experiments (both offline and online) on standard benchmark datasets, showing that meta-optimization chooses too small a learning rate by multiple orders of magnitude, even when run with a moderately long time horizon (100 steps) typical of work in the area. We believe short-horizon bias is a fundamental problem that needs to be addressed if meta-optimization is to scale to practical neural net training regimes.

## 1 INTRODUCTION

The learning rate is one of the most important and frustrating hyperparameters to tune in deep learning. Too small a value causes slow progress, while too large a value causes fluctuations or even divergence. While a fixed learning rate often works well for simpler problems, good performance on the ImageNet (Russakovsky et al., 2015) benchmark requires a carefully tuned schedule. A variety of decay schedules have been proposed for different architectures, including polynomial, exponential, staircase, etc. Learning rate decay is also required to achieve convergence guarantee for stochastic gradient methods under certain conditions (Bottou, 1998). Clever learning rate heuristics have resulted in large improvements in training efficiency (Goyal et al., 2017; Smith, 2017). A related hyperparameter is momentum; typically fixed to a reasonable value such as 0.9, careful tuning can also give significant performance gains (Sutskever et al., 2013). While optimizers such as Adam (Kingma & Ba, 2015) are often described as adapting coordinate-specific learning rates, in fact they also have global learning rate and momentum hyperparameters analogously to SGD, and tuning at least the learning rate can be important to good performance.

In light of this, it is not surprising that there have been many attempts to adapt learning rates, either online during optimization (Schraudolph, 1999; Schaul et al., 2013), or offline by fitting a learning rate schedule (Maclaurin et al., 2015). More ambitiously, others have attempted to learn an optimizer (Andrychowicz et al., 2016; Li & Malik, 2017; Finn et al., 2017; Lv et al., 2017; Wichrowska et al., 2017; Metz et al., 2017). All of these approaches are forms of meta-optimization, where one defines a meta-objective (typically the expected loss after some number of optimization steps) and tunes the hyperparameters to minimize this meta-objective. But because gradient-based meta-optimization can require thousands of updates, each of which unrolls the entire base-level optimization procedure, the meta-optimization is thousands of times more expensive than the base-level optimization. Therefore, the meta-objective must be defined with a much smaller time horizon

---

[*]Equal contribution.

Code available at `https://github.com/renmengye/meta-optim-public`

(e.g. hundreds of updates) than we are ordinarily interested in for large-scale optimization. The hope is that the learned hyperparameters or optimizer will generalize well to much longer time horizons. Unfortunately, we show that this is not achieved in this paper. This is because of a strong tradeoff between short-term and long-term performance, which we refer to as *short-horizon bias*.

In this work, we investigate the short-horizon bias both mathematically and empirically. First, we analyze a quadratic cost function with noisy gradients based on Schaul et al. (2013). We consider this a good proxy for neural net training because second-order optimization algorithms have been shown to train neural networks in orders-of-magnitude fewer iterations (Martens, 2010), suggesting that much of the difficulty of SGD training can be explained by quadratic approximations to the cost. In our noisy quadratic problem, the dynamics of SGD with momentum can be analyzed exactly, allowing us to derive the greedy-optimal (i.e. 1-step horizon) learning rate and momentum in closed form, as well as to (locally) minimize the long-horizon loss using gradient descent. We analyze the differences between the short-horizon and long-horizon schedules.

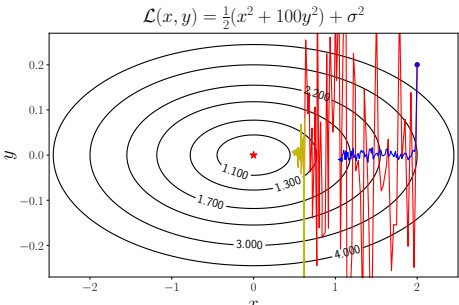

**Figure 1:** Aggressive learning rate (red) followed by a decay schedule (yellow) wins over conservative learning rate (blue) by making more progress along the low curvature direction ($x$ direction).

Interestingly, when the noisy quadratic problem is *either* deterministic or spherical, greedy schedules are optimal. However, when the problem is *both* stochastic and badly conditioned (as is most neural net training), the greedy schedules decay the learning rate far too quickly, leading to slow convergence towards the optimum. This is because reducing the learning rate dampens the fluctuations along high curvature directions, giving it a large immediate reduction in loss. But this comes at the expense of long-run performance, because the optimizer fails to make progress along low curvature directions. This phenomenon is illustrated in Figure 1, a noisy quadratic problem in 2 dimensions, in which two learning rate schedule are compared: a small fixed learning rate (blue), versus a larger fixed learning rate (red) followed by exponential decay (yellow). The latter schedule initially has higher loss, but it makes more progress towards the optimum, such that it achieves an even smaller loss once the learning rate is decayed.

Figure 2 shows this effect quantitatively for a noisy quadratic problem in 1000 dimensions (defined in Section 2.3). The solid lines show the loss after various numbers of steps of lookahead with a fixed learning rate; if this is used as the meta-objective, it favors small learning rates. The dashed curves show the loss if the same trajectories are followed by 50 steps with an exponentially decayed learning rate; these curves favor higher learning rates, and bear little obvious relationship to the solid ones. This illustrates the difficulty of selecting learning rates based on short-horizon information.

The second part of our paper empirically investigates gradient-based meta-optimization for neural net training. We consider two idealized meta-optimization algorithms: an offline algorithm which fits a learning rate decay schedule by running optimization many times from scratch, and an online algorithm which adapts the learning rate during training. Since our interest is in studying the effect of the meta-objective itself rather than failures of meta-optimization, we give the meta-optimizers sufficient time to optimize their meta-objectives well. We show that short-horizon meta-optimizers, both online and offline, dramatically underperform a hand-tuned fixed learning rate, and sometimes cause the base-level optimization progress to slow to a crawl, even with moderately long time horizons (e.g. 100 or 1000 steps) similar to those used in prior work on gradient-based meta-optimization.

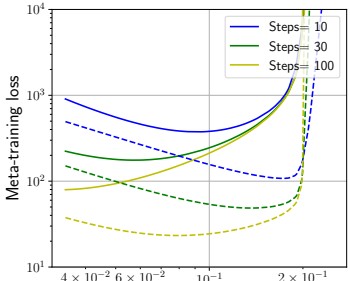

**Figure 2:** Short-horizon meta-objectives for the noisy quadratic problem. Solid: loss after $k$ updates with fixed learning rate. Dashed: loss after $k$ updates with fixed learning rate, followed by exponential decay.

In short, we expect that any meta-objective which does not correct for short-horizon bias will probably fail when run for a much longer time horizon than it was trained on. There are applications

where short-horizon meta-optimization is directly useful, such as few-shot learning (Santoro et al., 2016; Ravi & Larochelle, 2017). In those settings, short-horizon bias is by definition not an issue. But much of the appeal of meta-optimization comes from the possibility of using it to speed up or simplify the training of large neural networks. In such settings, short-horizon bias is a fundamental obstacle that must be addressed for meta-optimization to be practically useful.

## 2 NOISY QUADRATIC PROBLEM

In this section, we consider a toy problem which demonstrates the short-horizon bias and can be analyzed analytically. In particular, we borrow the noisy quadratic model of Schaul et al. (2013); the true function being optimized is a quadratic, but in each iteration we observe a noisy version with the correct curvature but a perturbed minimum. This can be equivalently viewed as noisy observations of the gradient, which are intended to capture the stochasticity of a mini-batch-based optimizer. We analyze the dynamics of SGD with momentum on this example, and compare the long-horizon-optimized and greedy-optimal learning rate schedules.

### 2.1 BACKGROUND

Approximating the cost surface of a neural network with a quadratic function has led to powerful insights and algorithms. Second-order optimization methods such as Newton-Raphson and natural gradient (Amari, 1998) iteratively minimize a quadratic approximation to the cost function. Hessian-free (H-F) optimization (Martens, 2010) is an approximate natural gradient method which tries to minimize a quadratic approximation using conjugate gradient. It can often fit deep neural networks in orders-of-magnitude fewer updates than SGD, suggesting that much of the difficulty of neural net optimization is captured by quadratic models. In the setting of Bayesian neural networks, quadratic approximations to the log-likelihood motivated the Laplace approximation (MacKay, 1992) and variational inference (Graves, 2011; Zhang et al., 2017). Koh & Liang (2017) used quadratic approximations to analyze the sensitivity of a neural network's predictions to particular training labels, thereby yielding insight into adversarial examples.

Such quadratic approximations to the cost function have also provided insights into learning rate and momentum adaptation. In a deterministic setting, under certain conditions, second-order optimization algorithms can be run with a learning rate of 1; for this reason, H-F was able to eliminate the need to tune learning rate or momentum hyperparameters. Martens & Grosse (2015) observed that for a deterministic quadratic cost function, greedily choosing the learning rate and momentum to minimize the error on the next step is equivalent to conjugate gradient (CG). Since CG achieves the minimum possible loss of any gradient-based optimizer on each iteration, the greedily chosen learning rates and momenta are optimal, in the sense that the greedy sequence achieves the minimum possible loss value of *any* sequence of learning rates and momenta. This property fails to hold in the stochastic setting, however, and as we show in this section, the greedy choice of learning rate and momentum can do considerably worse than optimal.

Our primary interest in this work is to adapt scalar learning rate and momentum hyperparameters shared across all dimensions. Some optimizers based on diagonal curvature approximations (Kingma & Ba, 2015) have been motivated in terms of adapting dimension-specific learning rates, but in practice, one still needs to tune scalar learning rate and momentum hyperparameters. Even K-FAC (Martens & Grosse, 2015), which is based on more powerful curvature approximations, has scalar learning rate and momentum hyperparameters. Our analysis applies to all of these methods since they can be viewed as performing SGD in a preconditioned space.

### 2.2 ANALYSIS

#### 2.2.1 NOTATIONS

We will primarily focus on the SGD with momentum algorithm in this paper. The update is written as follows:

$$\mathbf{v}^{(t+1)} = \mu^{(t)}\mathbf{v}^{(t)} - \alpha^{(t)}\nabla_{\boldsymbol{\theta}^{(t)}}\mathcal{L}, \tag{1}$$

$$\boldsymbol{\theta}^{(t+1)} = \boldsymbol{\theta}^{(t)} + \mathbf{v}^{(t+1)}, \tag{2}$$

where $\mathcal{L}$ is the loss function, $t$ is the training step, and $\alpha^{(t)}$ is the learning rate. We call the gradient trace $\mathbf{v}^{(t)}$ "velocity", and its decay constant $\mu^{(t)}$ "momentum". We denote the $i$th coordinate of a vector $\mathbf{v}$ as $v_i$. When we focus on a single dimension, we sometimes drop the dimension subscripts. We also denote $A(\cdot) = \mathbb{E}[\cdot]^2 + \mathbb{V}[\cdot]$, where $\mathbb{E}$ and $\mathbb{V}$ denote expectation and variance respectively.

### 2.2.2 PROBLEM FORMULATION

We now define the noisy quadratic model, where in each iteration, the optimizer is given the gradient for a noisy version of a quadratic cost function, where the curvature is correct but the minimum is sampled stochastically from a Gaussian distribution. We assume WLOG that the Hessian is diagonal because SGD is a rotation invariant algorithm, and therefore the dynamics can be analyzed in a coordinate system corresponding to the eigenvectors of the Hessian. We make the further (nontrivial) assumption that the noise covariance is also diagonal.[1] Mathematically, the stochastic cost function is written as:

$$\hat{\mathcal{L}}(\boldsymbol{\theta}) = \frac{1}{2} \sum_i h_i(\theta_i - c_i)^2, \tag{3}$$

where $\mathbf{c}$ is the stochastic minimum, and each $c_i$ follows a Gaussian distribution with mean $\theta_i^*$ and variance $\sigma_i^2$. The expected loss is given by:

$$\mathcal{L}(\boldsymbol{\theta}) = \mathbb{E}\left[\hat{\mathcal{L}}(\boldsymbol{\theta})\right] = \frac{1}{2} \sum_i h_i\left((\theta_i - \theta_i^*)^2 + \sigma_i^2\right). \tag{4}$$

The optimum of $\mathcal{L}$ is given by $\boldsymbol{\theta}^* = \mathbb{E}[\mathbf{c}]$; we assume WLOG that $\boldsymbol{\theta}^* = \mathbf{0}$. The stochastic gradient is given by $\frac{\partial \hat{\mathcal{L}}}{\partial \theta_i} = h_i(\theta_i - c_i)$. Since the deterministic gradient is given by $\frac{\partial \mathcal{L}}{\partial \theta_i} = h_i \theta_i$, the stochastic gradient can be viewed as a noisy Gaussian observation of the deterministic gradient with variance $h_i^2 \sigma_i^2$. This interpretation motivates the use of this noisy quadratic problem as a model of SGD dynamics.

We treat the iterate $\boldsymbol{\theta}^{(t)}$ as a random variable (where the randomness comes from the sampled $\mathbf{c}$'s); the expected loss in each iteration is given by

$$\mathbb{E}\left[\mathcal{L}(\boldsymbol{\theta}^{(t)})\right] = \mathbb{E}\left[\frac{1}{2} \sum_i h_i\left((\theta_i^{(t)})^2 + \sigma_i^2\right)\right] \tag{5}$$

$$= \frac{1}{2} \sum_i h_i\left(\mathbb{E}\left[\theta_i^{(t)}\right]^2 + \mathbb{V}\left[\theta_i^{(t)}\right] + \sigma_i^2\right). \tag{6}$$

### 2.2.3 OPTIMIZED AND GREEDY-OPTIMAL SCHEDULES

We are interested in adapting a global learning rate $\alpha^{(t)}$ and a global momentum decay parameter $\mu^{(t)}$ for each time step $t$. We first derive a recursive formula for the mean and variance of the iterates at each step, and then analyze the greedy-optimal schedule for $\alpha^{(t)}$ and $\mu^{(t)}$.

Several observations allow us to compactly model the dynamics of SGD with momentum on the noisy quadratic model. First, $\mathbb{E}[\mathcal{L}(\boldsymbol{\theta}^{(t)})]$ can be expressed in terms of $\mathbb{E}[\theta_i]$ and $\mathbb{V}[\theta_i]$ using Eqn. 5. Second, due to the diagonality of the Hessian and the noise covariance matrix, each coordinate evolves independently of the others. Third, the means and variances of the parameters $\theta_i^{(t)}$ and the velocity $v_i^{(t)}$ are functions of those statistics at the previous step.

Because each dimension evolves independently, we now drop the dimension subscripts. Combining these observations, we model the dynamics of SGD with momentum as a *deterministic* recurrence relation with sufficient statistics $\mathbb{E}[\theta^{(t)}]$, $\mathbb{E}[v^{(t)}]$, $\mathbb{V}[\theta^{(t)}]$, $\mathbb{V}[v^{(t)}]$, and $\Sigma_{\theta,v}^{(t)} = \mathrm{Cov}(\theta^{(t)}, v^{(t)})$. The dynamics are as follows:

---

[1]This amounts to assuming that the Hessian and the noise covariance are codiagonalizable. One heuristic justification for this assumption in the context of neural net optimization is that under certain assumptions, the covariance of the gradients is proportional to the Fisher information matrix, which is close to the Hessian (Martens, 2014).

**Theorem 1** (Mean and variance dynamics). *The expectations of the parameter $\theta$ and the velocity $v$ are updated as,*

$$\mathbb{E}\left[v^{(t+1)}\right] = \mu^{(t)}\mathbb{E}\left[v^{(t)}\right] - (\alpha^{(t)}h)\mathbb{E}\left[\theta^{(t)}\right],$$

$$\mathbb{E}\left[\theta^{(t+1)}\right] = \mathbb{E}\left[\theta^{(t)}\right] + \mathbb{E}\left[v^{(t+1)}\right].$$

*The variances of the parameter $\theta$ and the velocity $v$ are updated as*

$$\mathbb{V}\left[v^{(t+1)}\right] = \left(\mu^{(t)}\right)^2\mathbb{V}\left[v^{(t)}\right] + \left(\alpha^{(t)}h\right)^2\mathbb{V}\left[\theta^{(t)}\right] - 2\mu^{(t)}\alpha^{(t)}h\Sigma_{\theta,v}^{(t)} + \left(\alpha^{(t)}h\sigma\right)^2,$$

$$\mathbb{V}\left[\theta^{(t+1)}\right] = \left(1 - 2\alpha^{(t)}h\right)\mathbb{V}\left[\theta^{(t)}\right] + \mathbb{V}\left[v^{(t+1)}\right] + 2\mu^{(t)}\Sigma_{\theta,v}^{(t)},$$

$$\Sigma_{\theta,v}^{(t+1)} = \mu^{(t)}\Sigma_{\theta,v}^{(t)} - \alpha^{(t)}h\mathbb{V}\left[\theta^{(t)}\right] + \mathbb{V}\left[v^{(t+1)}\right].$$

By applying Theorem 1 recursively, we can obtain $\mathbb{E}[\boldsymbol{\theta}^{(t)}]$ and $\mathbb{V}[\boldsymbol{\theta}^{(t)}]$, and hence $\mathbb{E}[\mathcal{L}(\boldsymbol{\theta}^{(t)})]$, for every $t$. Therefore, using gradient-based optimization, we can fit a locally optimal learning rate and momentum schedule, i.e. a sequence of values $\{(\alpha^{(t)}, \mu^{(t)})\}_{t=1}^{T}$ which locally minimizes $\mathbb{E}[\mathcal{L}(\boldsymbol{\theta}^{(t)})]$ at some particular time $T$. We refer to this as the *optimized* schedule.

Furthermore, there is a closed-form solution for one-step lookahead, i.e., we can solve for the optimal learning rate $\alpha^{(t)*}$ and momentum $\mu^{(t)*}$ that minimizes $\mathbb{E}[\mathcal{L}(\boldsymbol{\theta}^{(t+1)})]$ given the statistics at time $t$. We call this as the *greedy-optimal* schedule.

**Theorem 2** (Greedy-optimal learning rate and momentum). *The greedy-optimal learning rate and momentum schedule is given by*

$$\alpha^{(t)*} = \frac{\sum_i h_i^2 A\left(\theta_i^{(t)}\right)\left[\sum_j h_j A\left(v_j^{(t)}\right)\right] - \left(\sum_j h_j\mathbb{E}\left[\theta_j^{(t)}v_j^{(t)}\right]\right)h_i^2\,\mathbb{E}\left[\theta_i^{(t)}v_i^{(t)}\right]}{\sum_i h_i^3\left[A\left(\theta_i^{(t)}\right) + \sigma_i^2\right]\left[\sum_j h_j A\left(v_j^{(t)}\right)\right] - \left(\sum_j h_j^2\,\mathbb{E}\left[\theta_j^{(t)}v_j^{(t)}\right]\right)h_i^2\,\mathbb{E}\left[\theta_i^{(t)}v_i^{(t)}\right]},$$

$$\mu^{(t)*} = -\frac{\sum_i h_i\left(1 - \alpha^{(t)*}h_i\right)\mathbb{E}\left[\theta_i^{(t)}v_i^{(t)}\right]}{\sum_i h_i A\left(v_i^{(t)}\right)}.$$

Note that Schaul et al. (2013) derived the greedy optimal learning rate for SGD, and Theorem 2 extends it to the greedy optimal learning rate and momentum for SGD with momentum.

### 2.2.4 UNIVARIATE AND SPHERICAL CASES

As noted in Section 2.1, Martens & Grosse (2015) found the greedy choice of $\alpha$ and $\mu$ to be optimal for gradient descent on deterministic quadratic objectives. We now show that the greedy schedule is also optimal for SGD *without* momentum in the case of univariate noisy quadratics, and hence also for multivariate ones with spherical Hessians and gradient covariances. In particular, the following holds for SGD without momentum on a univariate noisy quadratic:

**Theorem 3** (Optimal learning rate, univariate). *For all $T \in \mathbb{N}$, the sequence of learning rates $\{\alpha^{(t)*}\}_{t=1}^{T-1}$ that minimizes $\mathcal{L}(\theta^{(T)})$ is given by*

$$\alpha^{(t)*} = \frac{A\left(\theta^{(t)}\right)}{h\left(A\left(\theta^{(t)}\right) + \sigma^2\right)}. \tag{7}$$

*Moreover, this agrees with the greedy-optimal learning rate schedule as derived by Schaul et al. (2013).*

If the Hessian and the gradient covariance are both spherical, then each dimension evolves identically and independently according to the univariate dynamics. Of course, one is unlikely to encounter an optimization problem where both are exactly spherical. But some approximate second-order optimizers, such as K-FAC, can be viewed as preconditioned SGD, i.e. SGD in a transformed

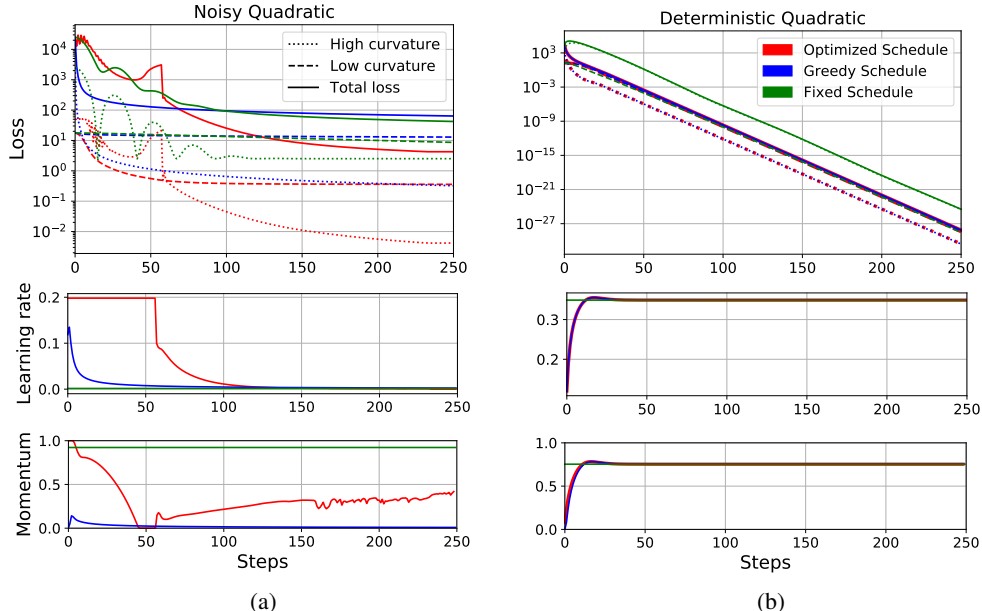

**Figure 3:** Comparisons of the optimized learning rates and momenta trained by gradient descent (red), greedy learning rates and momenta (blue), and the optimized fixed learning rate and momentum (green) in both noisy (a) and deterministic (b) quadratic settings. In the deterministic case, our optimized schedule matched the greedy one, just as the theory predicts.

space where the Hessian and the gradient covariance are better conditioned (Martens & Grosse, 2015). In principle, with a good enough preconditioner, the Hessian and the gradient covariance would be close enough to spherical that a greedy choice of $\alpha$ and $\mu$ would perform well. It will be interesting to investigate whether any practical optimization algorithms demonstrate this behavior.

## 2.3 EXPERIMENTS

In this section, we compare the optimized and greedy-optimal schedules on a noisy quadratic problem. We chose a 1000 dimensional quadratic cost function with the curvature distribution from Li (2005), on which CG achieves its worst-case convergence rate. We assume that $h_i = \mathbb{V}[\frac{\partial \mathcal{L}}{\partial \theta_i}]$, and hence $\sigma_i^2 = \frac{1}{h_i}$; this choice is motivated by the observations that under certain assumptions, the Fisher information matrix is a good approximation to the Hessian matrix, but also reflects the covariance structure of the gradient noise (Martens, 2014). We computed the greedy-optimal schedules using Theorem 3. For the optimized schedules, we minimized the expected loss at time $T = 250$ using Adam using Adam (Kingma & Ba, 2015), with a learning rate 0.003 and 500 steps. We set an upper bound for the learning rate which prevented the loss component for any dimension from becoming larger than its initial value; this was needed because otherwise the optimized schedule allowed the loss to temporarily grow very large, a pathological solution which would be unstable on realistic problems. We also considered fixed learning rate and momentum, with the two hyperparameters fit using Adam. The training curves and the corresponding learning rates and momenta are shown in Figure 3(a). The optimized schedule achieved a much lower final expected loss value (4.25) than was obtained by the greedy-optimal schedule (63.86) or fixed schedule (42.19).

We also show the sums of the losses along the 50 highest curvature directions and 50 lowest curvature directions. We find that under the optimized schedule, the losses along the high curvature directions hardly decrease initially. However, because it maintains a high learning rate, the losses along the low curvature directions decrease significantly. After 50 iterations, it begins decaying the learning rate, at which point it achieves a large drop in both the high-curvature and total losses. On the other hand, under the greedy-optimal schedule, the learning rates and momenta become small very early on, which immediately reduces the losses on the high curvature directions, and hence also the total loss. However, in the long term, since the learning rates are too small to make substantial progress along the low curvature directions, the total loss converged to a much higher value in the

end. This gives valuable insight into the nature of the short-horizon bias in meta-optimization: short-horizon objectives will often encourage the learning rate and momentum to decay quickly, so as to achieve the largest gain in the short term, but at the expense of long-run performance.

It is interesting to compare this behavior with the deterministic case. We repeated the above experiment for a *deterministic* quadratic cost function (i.e. $\sigma_i^2 = 0$) with the same Hessian; results are shown in Figure 3(b). The greedy schedule matches the optimized one, as predicted by the analysis of Martens & Grosse (2015). This result illustrates that stochasticity is necessary for short-horizon bias to manifest. Interestingly, the learning rate and momentum schedules in the deterministic case are nearly flat, while the optimized schedules for the stochastic case are much more complex, suggesting that stochastic optimization raises a different set of issues for hyperparameter adaptation.

## 3 GRADIENT-BASED META-OPTIMIZATION

We now turn our attention to gradient-based hyperparameter optimization. A variety of approaches have been proposed which tune hyperparameters by doing gradient descent on a meta-objective (Schraudolph, 1999; Maclaurin et al., 2015; Andrychowicz et al., 2016). We empirically analyze an idealized version of a gradient-based meta-optimization algorithm called stochastic meta-descent (SMD) (Schraudolph, 1999). Our version of SMD is idealized in two ways: first, we drop the algorithmic tricks used in prior work, and instead allow the meta-optimizer more memory and computation than would be economical in practice. Second, we limit the representational power of our meta-model: whereas Andrychowicz et al. (2016) aimed to learn a full optimization algorithm, we focus on the much simpler problem of adapting learning rate and momentum hyperparameters, or schedules thereof. The aim of these two simplifications is that we would like to do a good enough job of optimizing the meta-objective that any base-level optimization failures can be attributed to deficiencies in the meta-objective itself (such as short-horizon bias) rather than incomplete meta-optimization.

Despite these simplifications, we believe our experiments are relevant to practical meta-optimization algorithms which optimize the meta-objective less thoroughly. Since the goal of the meta-optimizer is to adapt two hyperparameters, it's possible that poor meta-optimization could cause the hyperparameters to get stuck in regions that happen to perform well; indeed, we observed this phenomenon in some of our early explorations. But it would be dangerous to rely on poor meta-optimization, since improved meta-optimization methods would then lead to worse base-level performance, and tuning the meta-optimizer could become a roundabout way of tuning learning rates and momenta.

We also believe our experiments are relevant to meta-optimization methods which aim to learn entire algorithms. Even if the learned algorithms don't have explicit learning rate parameters, it's possible for a learning rate schedule to be encoded into an algorithm itself; for instance, Adagrad (Duchi et al., 2011) implicitly uses a polynomial decay schedule because it sums rather than averages the squared derivatives in the denominator. Hence, one would need to worry about whether the meta-optimizer is implicitly fitting a learning rate schedule that's optimized for short-term performance.

### 3.1 BACKGROUND: STOCHASTIC META-DESCENT

The high-level idea of stochastic meta-descent (SMD) (Schraudolph, 1999) is to perform gradient descent on the learning rate, or any other differentiable hyperparameters. This is feasible since any gradient based optimization algorithm can be unrolled as a computation graph (see Figure 4), and automatic differentiation is readily available in most deep learning libraries.

There are two basic types of automatic differentiation (autodiff) methods: *forward mode* and *reverse mode*. In forward mode autodiff, directional derivatives are computed alongside the forward computation. In contrast, reverse mode autodiff (a.k.a. backpropagation) computes the gradients moving backwards through the computation graph. Meta-optimization using reverse mode can be computationally demanding due to memory constraints, since the parameters need to be stored at every step. Maclaurin et al. (2015) got around this by cleverly exploiting approximate reversibility to minimize the memory cost of activations. Since we are optimizing only two hyperparameters, however, forward mode autodiff can be done cheaply. Here, we provide the forward differentiation

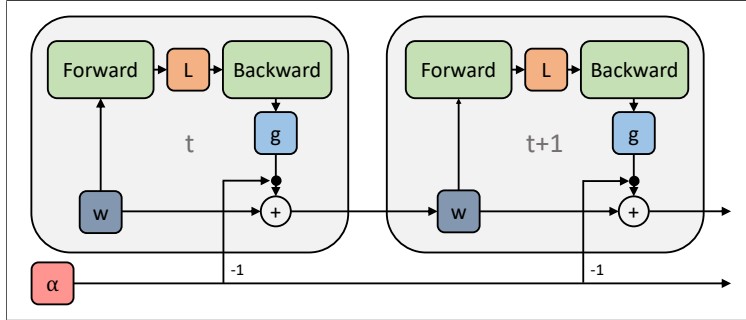

**Figure 4:** Regular SGD in the form of a computation graph. The learning rate parameter $\alpha$ is part of the differentiable computations.

equations for obtaining the gradient of vanilla SGD learning rate. Let $\frac{d\boldsymbol{\theta}_t}{d\alpha}$ be $\boldsymbol{u}_t$, and $\frac{d\mathcal{L}_t}{d\alpha}$ be $\alpha'$, and the Hessian at step $t$ to be $H_t$. By chain rule, we get,

$$\alpha' = \boldsymbol{g}_t \cdot \boldsymbol{u}_{t-1}, \tag{8}$$

$$\boldsymbol{u}_t = \boldsymbol{u}_{t-1} - \boldsymbol{g}_t - \alpha H_t \boldsymbol{u}_{t-1}. \tag{9}$$

While the Hessian is infeasible to construct explicitly, the Hessian-vector product in Equation 9 can be computed efficiently using reverse-on-reverse (Werbos, 1988) or forward-on-reverse automatic differentiation (Pearlmutter, 1994), in time linear in the cost of the forward pass. See Schraudolph (2002) for more details.

Using the gradients with respect to hyper-parameters, as given in Eq. 9, we can apply gradient based meta-optimization, just like optimizing regular parameters. It is worth noting that, although SMD was originally proposed for optimizing vanilla SGD, in practice it can be applied to other optimization algorithms such as SGD with momentum or Adam (Kingma & Ba, 2015). Moreover, gradient-based optimizers other than SGD can be used for the meta-optimization as well.

The basic SMD algorithm is given as Algorithm 1. Here, $\alpha$ is a set of hyperparameters (e.g. learning rate), and $\alpha_0$ are inital hyperparameter values; $\boldsymbol{\theta}$ is a set of optimization intermediate variables, such as weights and velocities; $\eta$ is a set of meta-optimizer hyperparameters (e.g. meta learning rate). $\texttt{BGrad}(y, x, dy)$ is the backward gradient function that computes the gradients of the loss function wrt. $\boldsymbol{\theta}$, and $\texttt{FGrad}(y, x, dx)$ is the forward gradient function that accumulates the gradients of $\boldsymbol{\theta}$ with respect to $\alpha$. $\texttt{Step}$ and $\texttt{MetaStep}$ optimize regular parameters and hyperparameters, respectively, for one step using gradient-based methods. Additionally, $T$ is the lookahead window size, and $M$ is the number of meta updates.

---

**Algorithm 1:** Stochastic Meta-Descent

**Input:** $\alpha_0, \eta, \boldsymbol{\theta}, T, M$
**Output:** $\alpha$
$\boldsymbol{\theta}_0 \leftarrow \boldsymbol{\theta}$;
$\alpha \leftarrow \alpha_0$;
**for** $m \leftarrow 1 \dots M$ **do**
 $\boldsymbol{u} \leftarrow \boldsymbol{0}$;
 **for** $t \leftarrow 1 \dots T$ **do**
  $X, \boldsymbol{y} \leftarrow \texttt{GetMiniBatch()}$;
  $\boldsymbol{g} \leftarrow \texttt{BGrad}(L(X, \boldsymbol{y}, \boldsymbol{\theta}), \boldsymbol{\theta}, 1)$;
  $\boldsymbol{\theta}_{new} \leftarrow \texttt{Step}(\boldsymbol{\theta}, \boldsymbol{g}, \alpha)$;
  $\alpha' \leftarrow \boldsymbol{g} \cdot \boldsymbol{u}$;
  $\boldsymbol{u} \leftarrow \texttt{FGrad}(\boldsymbol{\theta}_{new}, [\alpha, \boldsymbol{\theta}], [1, \boldsymbol{u}])$;
  $\boldsymbol{\theta} \leftarrow \boldsymbol{\theta}_{new}$;
 $\alpha \leftarrow \texttt{MetaStep}(\alpha, \alpha', \eta)$;
 $\boldsymbol{\theta} \leftarrow \boldsymbol{\theta}_0$
**return** $\alpha$

---

**Simplifications from the original SMD algorithm.** The original SMD algorithm (Schraudolph, 1999) fit coordinate-wise adaptive learning rates with intermediate gradients ($\boldsymbol{u}_t$) accumulated throughout the process of training. Since computing separate directional derivatives for each coordinate using forward mode autodiff is computationally prohibitive, the algorithm used approximate updates. Both features introduced bias into the meta-gradients. We make several changes to the original algorithm. First, we tune only a global learning rate parameter. Second, we use exact forward mode accumulation because this is feasible for a single learning rate. Third, rather than accumulate directional derivatives during training, we compute the meta-updates on separate SGD

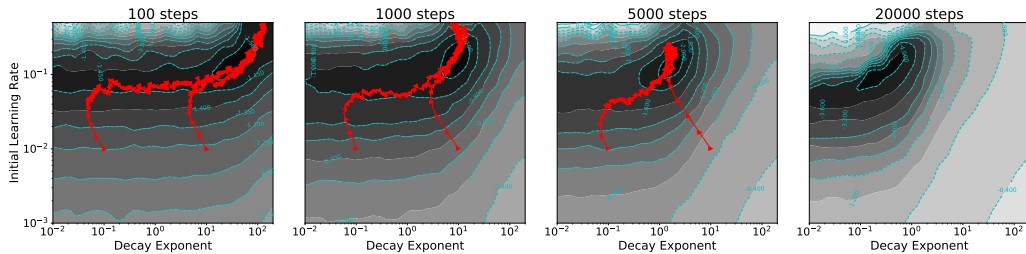

**Figure 5:** Meta-objective surfaces and SMD trajectories (red) optimizing initial effective learning rate and decay exponent with horizons of {100, 1k, 5k, 20k} steps[2]. 2.5k random samples with Gaussian interpolation are used to illustrate the meta-objective surface.

trajectories simulated using fixed network parameters. Finally, we compute multiple meta-updates in order to ensure that the meta-objective is optimized sufficiently well. Together, these changes ensure unbiased meta-gradients, as well as careful optimization of the meta-objective, at the cost of high computational overhead. We do not recommend this approach as a practical SMD implementation, but rather as a way of understanding the biases in the meta-objective itself.

## 3.2 OFFLINE META-OPTIMIZATION

To understand the sensitivity of the optimized hyperparameters to the horizon, we first carried out an offline experiment on a multi-layered perceptron (MLP) on MNIST (LeCun et al., 1998). Specifically, we fit learning rate decay schedules offline by repeatedly training the network, and a single meta-gradient was obtained from each training run.

**Learnable decay schedule.** We used a parametric learning rate decay schedule known as *inverse time decay* (Welling & Teh, 2011): $\alpha_t = \frac{\alpha_0}{(1+\frac{t}{K})^\beta}$, where $\alpha_0$ is the initial learning rate, $t$ is the number of training steps, $\beta$ is the learning rate decay exponent, and $K$ is the time constant. We jointly optimized $\alpha_0$ and $\beta$. We fixed $\mu = 0.9$, $K = 5000$ for simplicity.

**Experimental details.** The network had two layers of 100 hidden units, with ReLU activations. Weights were initialized with a zero-mean Gaussian with standard deviation 0.1. We used a warm start from a network trained for 50 SGD with momentum steps, using $\alpha = 0.1, \mu = 0.9$. (We used a warm start because the dynamics are generally different at the very start of training.) For SMD optimization, we trained all hyperparameters in log space using Adam optimizer, with 5k meta steps.

Figure 5 shows SMD optimization trajectories on the meta-objective surfaces, initialized with multiple random hyperparameter settings. The SMD trajectories appear to have converged to the global optimum.

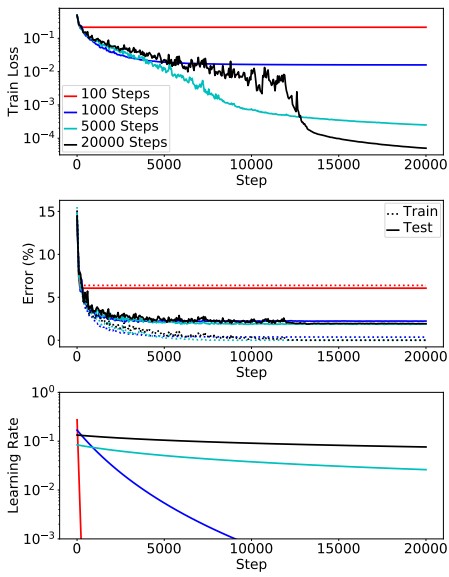

**Figure 6:** Training curves with best learning rate schedules from meta-objective surfaces with {100, 1k, 5k, 20k} step horizons.

Importantly, the meta-objectives with longer horizons favored a much smaller learning rate decay exponent $\beta$, leading to a more gradual decay schedule. The meta-objective surfaces were very different depending on the time horizon, and the final $\beta$ value differed by over two orders of magnitude between 100 and 20k step horizons.

We picked the best learning rate schedules from meta-objective surfaces (in Figure 5), and obtained the training curves of a network shown in Figure 6. The resulting training loss at 20k steps with

---

[2]We encountered some optimization difficulties for SMD with horizon of 20k steps. Since those are not the focus of this paper, we left out the trajectories of 20k steps to avoid confusions.

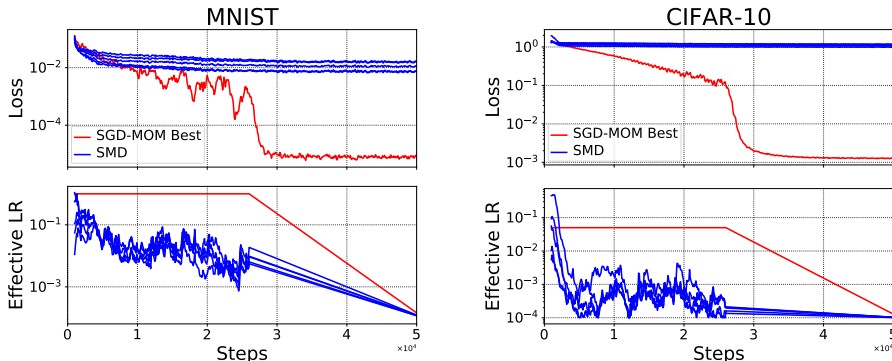

**Figure 7:** Training curves and learning rates from online SMD with lookahead of 5 steps (blue), and hand-tuned fixed learning rate (red). Each blue curve corresponds to a different initial learning rate.

the 100 step horizon was over *three* orders of magnitude larger than with the 20k step horizon. In general, short horizons gave better performance initially, but were surpassed by longer horizons. The differences in *error* were less drastic, but we see that the 100 step network was severely undertrained, and the 1k step network achieved noticeably worse test error than the longer-horizon ones.

### 3.3 ONLINE META-OPTIMIZATION

In this section, we study whether online adaptation also suffers from short-horizon bias. Specifically, we used Algorithm 1) to adapt the learning rate and momentum hyperparameters online while a network is trained. We experimented with an MLP on MNIST and a CNN on CIFAR-10 (Krizhevsky, 2009).

**Experimental details.** For the MNIST experiments, we used an MLP network with two hidden layers of 100 units, with ReLU activations. Weights were initialized with a zero-mean Gaussian with standard deviation 0.1. For CIFAR-10 experiments, we used a CNN network adapted from Caffe (Jia et al., 2014), with 3 convolutional layers of filter size $3 \times 3$ and depth [32, 32, 64], and $2 \times 2$ max pooling with stride 2 after every convolution layer, and follwed by a fully connected hidden layer of 100 units. Meta-optimization was done with 100 steps of Adam for every 10 steps of regular training. We adapted the learning rate $\alpha$ and momentum $\mu$. After 25k steps, adaptation was stopped, and we trained for another 25k steps with an exponentially decaying learning rate such that it reached 1e-4 on the last time step. We re-parameterized the learning rate with the effective learning rate $\alpha_{\text{eff}} = \frac{\alpha}{1-\mu}$, and the momentum with $1 - \mu$, so that they can be optimized more smoothly in the log space.

Figure 7 shows training curves both with online SMD and with hand-tuned fixed learning rate and momentum hyperparameters. We show several SMD runs initialized from widely varying hyperparameters; all the SMD runs behaved similarly, suggesting it optimized the meta-objective efficiently enough. Under SMD, learning rates were quickly decreased to very small values, leading to slow progress in the long term, consistent with the noisy quadratic and offline adaptation experiments.

As online SMD can be too conservative in the choice of learning rate, it is natural to ask whether removing the stochasticity in the lookahead sequence can fix the problem. We therefore considered online SMD where the entire lookahead trajectory used a *single* mini-batch, hence removing the stochasticity. As shown in Figure 8, this deterministic lookahead scheme led to the opposite problem: the adapted learning rates were very large, leading to instability. We conclude that the stochasticity of mini-batch training cannot be simply ignored in meta-optimization.

## 4 CONCLUSION

In this paper, we analyzed the problem of short-horizon bias in meta-optimization. We presented a noisy quadratic toy problem which we analyzed mathematically, and observed that the optimal

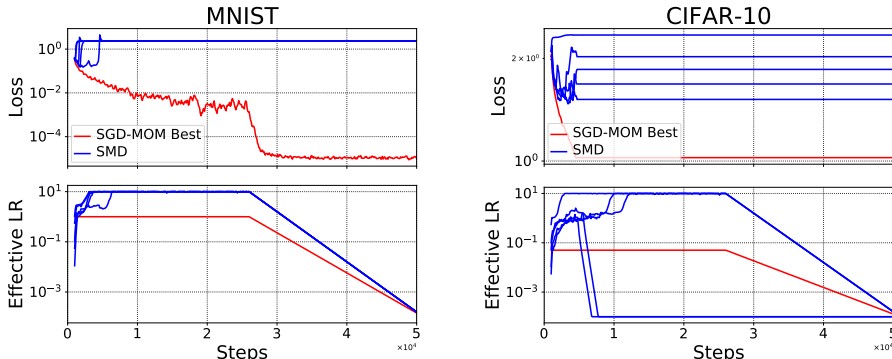

**Figure 8:** Online SMD with deterministic lookahead of 5 steps (blue), compared with a manually tuned fixed learning rate (red). Other settings are the same as Figure 7.

learning rate schedule differs greatly from a greedy schedule that minimizes training loss one step ahead. While the greedy schedule tends to decay the learning rate drastically to reduce the loss on high curvature directions, the optimal schedule keeps a high learning rate in order to make steady progress on low curvature directions, and eventually achieves far lower loss. We showed that this bias stems from the combination of stochasticity and ill-conditioning: when the problem is *either* deterministic or spherical, the greedy learning rate schedule is globally optimal; however, when the problem is both stochastic and ill-conditioned (as is most neural net training), the greedy schedule performs poorly. We empirically verified the short-horizon bias in the context of neural net training by applying gradient based meta-optimization, both offline and online. We found the same pathological behaviors as in the noisy quadratic problem — a fast learning rate decay and poor long-run performance.

While our results suggest that meta-optimization should not be applied blindly, our noisy quadratic analysis also provides grounds for optimism: by removing ill-conditioning (by using a good preconditioner) and/or stochasticity (with large batch sizes or variance reduction techniques), it may be possible to enter the regime where short-horizon meta-optimization works well. It remains to be seen whether this is achievable with existing optimization algorithms.

**Acknowledgement**    YW is supported by a Google PhD Fellowship. RL is supported by Connaught International Scholarships.

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

## A  PROOFS OF THEOREMS

The proofs are organized as follows; we provide a proof to Theorem 1 in A.1, a proof to Theorem 2 in A.2 and a proof to Theorem 3 in A.3.

### A.1  MODEL DYNAMICS

Recall the stochastic gradient descent with momentum is defined as follows,

$$v^{(t+1)} = \mu^{(t)}v^{(t)} - \alpha^{(t)}(h\theta^{(t)} + h\sigma\xi), \quad \xi \sim \mathcal{N}(0,1)$$
$$\theta^{(t+1)} = \theta^{(t)} + v^{(t+1)} = \theta^{(t)} + \mu^{(t)}v^{(t)} - \alpha^{(t)}(h\theta^{(t)} + h\sigma\xi)$$
$$= (1 - \alpha^{(t)}h)\theta^{(t)} + \mu^{(t)}v^{(t)} + h\sigma\xi.$$

#### A.1.1  DYNAMICS OF THE EXPECTATION

We calculate the mean of the velocity $v^{(t+1)}$,

$$\mathbb{E}\left[v^{(t+1)}\right] = \mathbb{E}\left[\mu^{(t)}v^{(t)} - \alpha^{(t)}h\theta^{(t)}\right]$$
$$= \mu^{(t)}\mathbb{E}\left[v^{(t)}\right] - \alpha^{(t)}h\mathbb{E}\left[\theta^{(t)}\right]. \tag{10}$$

We calculate the mean of the parameter $\theta^{(t+1)}$,

$$\mathbb{E}\left[\theta^{(t+1)}\right] = \mathbb{E}\left[\theta^{(t)}\right] + \mathbb{E}\left[v^{(t+1)}\right]. \tag{11}$$

Let's assume the following initial conditions:

$$\mathbb{E}\left[v^{(0)}\right] = 0$$
$$\mathbb{E}\left[\theta^{(0)}\right] = E_0.$$

Then Eq.(10) and Eq.(11) describes how $\mathbb{E}\left[\theta^{(t)}\right], \mathbb{E}\left[v^{(t)}\right]$ changes over time $t$.

### A.1.2 DYNAMICS OF THE VARIANCE

We calculate the variance of the velocity $v^{(t+1)}$,

$$\mathbb{V}\left[v^{(t+1)}\right] = \mathbb{V}\left[\mu^{(t)}v^{(t)} - \alpha^{(t)}h\theta^{(t)}\right] + (\alpha^{(t)}h\sigma)^2$$
$$= (\mu^{(t)})^2\mathbb{V}\left[v^{(t)}\right] + (\alpha^{(t)}h)^2\mathbb{V}\left[\theta^{(t)}\right] - 2\mu^{(t)}\alpha^{(t)}h \cdot \text{Cov}\left(\theta^{(t)}, v^{(t)}\right) + (\alpha^{(t)}h\sigma)^2. \tag{12}$$

The variance of the parameter $\theta^{(t+1)}$ is given by,

$$\mathbb{V}\left[\theta^{(t+1)}\right] = \mathbb{V}\left[\theta^{(t)}\right] + \mathbb{V}\left[v^{(t+1)}\right] + 2\left(\mu^{(t)}\text{Cov}\left(\theta^{(t)}, v^{(t)}\right) - \alpha^{(t)}h\mathbb{V}\left[\theta^{(t)}\right]\right). \tag{13}$$

We also need to derive how the covariance of $\theta$ and $v$ changes over time:

$$\text{Cov}\left(\theta^{(t+1)}, v^{(t+1)}\right) = \text{Cov}\left((\theta^{(t)} + v^{(t+1)}), v^{(t+1)}\right)$$
$$= \text{Cov}\left(\theta^{(t)}, v^{(t+1)}\right) + \mathbb{V}\left[v^{(t+1)}\right]$$
$$= \mu^{(t)}\text{Cov}\left(\theta^{(t)}, v^{(t)}\right) - \alpha^{(t)}h\mathbb{V}\left[\theta^{(t)}\right] + \mathbb{V}\left[v^{(t+1)}\right]. \tag{14}$$

Let's assume the following initial conditions:

$$\mathbb{V}\left[v^{(0)}\right] = 0$$
$$\mathbb{V}\left[\theta^{(0)}\right] = V_0$$
$$\text{Cov}\left(\theta^{(0)}, v^{(0)}\right) = 0.$$

Combining Eq.(12-14), we obtain the following dynamics (from $t = 0, \ldots, T - 1$):

$$\mathbb{V}\left[v^{(t+1)}\right] = (\mu^{(t)})^2\mathbb{V}\left[v^{(t)}\right] + (\alpha^{(t)}h)^2\mathbb{V}\left[\theta^{(t)}\right] - 2\mu^{(t)}\alpha^{(t)}h \cdot \text{Cov}\left(\theta^{(t)}, v^{(t)}\right) + (\alpha^{(t)}h\sigma)^2$$
$$\mathbb{V}\left[\theta^{(t+1)}\right] = \mathbb{V}\left[\theta^{(t)}\right] + \mathbb{V}\left[v^{(t+1)}\right] + 2\left(\mu^{(t)}\text{Cov}\left(\theta^{(t)}, v^{(t)}\right) - \alpha^{(t)}h\mathbb{V}\left[\theta^{(t)}\right]\right)$$
$$\text{Cov}\left(\theta^{(t+1)}, v^{(t+1)}\right) = \mu^{(t)}\text{Cov}\left(\theta^{(t)}, v^{(t)}\right) - \alpha^{(t)}h\mathbb{V}\left[\theta^{(t)}\right] + \mathbb{V}\left[v^{(t+1)}\right].$$

## A.2 GREEDY OPTIMALITY

### A.2.1 UNIVARIATE CASE

The loss at time step $t$ is,

$$
\begin{aligned}
\mathcal{L}^{(t+1)} &= \frac{1}{2}h\left(\mathbb{E}\left[\theta^{(t+1)}\right]^2 + \mathbb{V}\left[\theta^{(t+1)}\right]\right) \\
&= \frac{1}{2}h\Big[\left(\mathbb{E}\left[\theta^{(t)}\right] + \mu^{(t)}\mathbb{E}\left[v^{(t)}\right] - (\alpha^{(t)}h)\mathbb{E}\left[\theta^{(t)}\right]\right)^2 + \mathbb{V}\left[\theta^{(t)}\right] + (\mu^{(t)})^2\mathbb{V}\left[v^{(t)}\right] + (\alpha^{(t)}h)^2\mathbb{V}\left[\theta^{(t)}\right] \\
&\quad - 2\mu^{(t)}\alpha^{(t)}h \cdot \mathrm{Cov}\left(\theta^{(t)}, v^{(t)}\right) + (\alpha^{(t)}h\sigma)^2 + 2\left(\mu^{(t)}\mathrm{Cov}\left(\theta^{(t)}, v^{(t)}\right) - \alpha^{(t)}h\mathbb{V}\left[\theta^{(t)}\right]\right)\Big] \\
&= \frac{1}{2}h\Big[\left((1 - \alpha^{(t)}h)\mathbb{E}\left[\theta^{(t)}\right] + \mu^{(t)}\mathbb{E}\left[v^{(t)}\right]\right)^2 + (1 - \alpha^{(t)}h)^2\mathbb{V}\left[\theta^{(t)}\right] + (\mu^{(t)})^2\mathbb{V}\left[v^{(t)}\right] \\
&\quad + 2\mu^{(t)}(1 - \alpha^{(t)}h)\mathrm{Cov}\left(\theta^{(t)}, v^{(t)}\right) + (\alpha^{(t)}h\sigma)^2\Big] \\
&= \frac{1}{2}h\Big[(1 - \alpha^{(t)}h)^2\left(\mathbb{E}\left[\theta^{(t)}\right]^2 + \mathbb{V}\left[\theta^{(t)}\right]\right) + (\mu^{(t)})^2\left(\mathbb{E}\left[v^{(t)}\right]^2 + \mathbb{V}\left[v^{(t)}\right]\right) \\
&\quad + 2\mu^{(t)}(1 - \alpha^{(t)}h)\left(\mathbb{E}\left[\theta^{(t)}\right]\mathbb{E}\left[v^{(t)}\right] + \mathrm{Cov}\left(\theta^{(t)}, v^{(t)}\right)\right) + (\alpha^{(t)}h\sigma)^2\Big].
\end{aligned}
$$

For simplicity, we denote $A(\cdot) = \mathbb{E}\left[\cdot\right]^2 + \mathbb{V}\left[\cdot\right]$, and notice that $\mathbb{E}\left[\theta^{(t)}v^{(t)}\right] = \mathbb{E}\left[\theta^{(t)}\right]\mathbb{E}\left[v^{(t)}\right] + \mathrm{Cov}\left(\theta^{(t)}, v^{(t)}\right)$, hence,

$$
\mathcal{L}^{(t+1)} = \frac{1}{2}h\Big[(1 - \alpha^{(t)}h)^2 A(\theta^{(t)}) + (\mu^{(t)})^2 A(v^{(t)}) + 2\mu^{(t)}(1 - \alpha^{(t)}h)\mathbb{E}\left[\theta^{(t)}v^{(t)}\right] + (\alpha^{(t)}h\sigma)^2\Big].
\tag{15}
$$

In order to find the optimal learning rate and momentum for minimizing $\mathcal{L}^{(t+1)}$, we take the derivative with respect to $\alpha^{(t)}$ and $\mu^{(t)}$, and set it to 0:

$$
\nabla_{\alpha^{(t)}}\mathcal{L}^{(t+1)} = (1 - \alpha^{(t)}h)A(\theta^{(t)}) \cdot (-h) - \mu^{(t)}h\mathbb{E}\left[\theta^{(t)}v^{(t)}\right] + \alpha^{(t)}(h\sigma)^2 = 0
$$

$$
\alpha^{(t)}h(A(\theta^{(t)}) + \sigma^2) = A(\theta^{(t)}) + \mu^{(t)}\mathbb{E}\left[\theta^{(t)}v^{(t)}\right]
$$

$$
\nabla_{\mu^{(t)}}\mathcal{L}^{(t+1)} = \mu^{(t)}A(v^{(t)}) + (1 - \alpha^{(t)}h)\mathbb{E}\left[\theta^{(t)}v^{(t)}\right] = 0
$$

$$
\mu^{(t)*} = -\frac{(1 - \alpha^{(t)}h)\mathbb{E}\left[\theta^{(t)}v^{(t)}\right]}{A(v^{(t)})}
$$

$$
\alpha^{(t)}h(A(\theta^{(t)}) + \sigma^2) = A(\theta^{(t)}) - \frac{(1 - \alpha^{(t)}h)\mathbb{E}\left[\theta^{(t)}v^{(t)}\right]}{A(v^{(t)})}\mathbb{E}\left[\theta^{(t)}v^{(t)}\right]
$$

$$
\alpha^{(t)*} = \frac{A(\theta^{(t)})A(v^{(t)}) - \mathbb{E}\left[\theta^{(t)}v^{(t)}\right]^2}{h\left(A(v^{(t)})(A(\theta^{(t)}) + \sigma^2) - \mathbb{E}\left[\theta^{(t)}v^{(t)}\right]^2\right)}.
$$

### A.2.2 HIGH DIMENSION CASE

The loss is the sum of losses along all directions:

$$
\mathcal{L}^{(t+1)} = \sum_i \frac{1}{2}h_i\Big[(1 - \alpha^{(t)}h_i)^2 A(\theta_i^{(t)}) + (\mu^{(t)})^2 A(v_i^{(t)}) + 2\mu^{(t)}(1 - \alpha^{(t)}h_i)\mathbb{E}\left[\theta_i^{(t)}v_i^{(t)}\right] + (\alpha^{(t)}h_i\sigma_i)^2\Big]
$$

Now we obtain optimal learning rate and momentum by setting the derivative to 0,

$$\nabla_{\alpha^{(t)}}\mathcal{L}^{(t+1)} = \sum_i h_i\Big[(1 - \alpha^{(t)}h_i)A(\theta_i^{(t)}) \cdot (-h_i) - \mu^{(t)}h_i\mathbb{E}\Big[\theta_i^{(t)}v_i^{(t)}\Big] + \alpha^{(t)}(h_i\sigma_i)^2\Big] = 0$$

$$\alpha^{(t)}\sum_i\Big((h_i)^3(A(\theta_i^{(t)}) + (\sigma_i)^2)\Big) = \sum_i\Big((h_i)^2A(\theta_i^{(t)}) + \mu^{(t)}(h_i)^2\mathbb{E}\Big[\theta_i^{(t)}v_i^{(t)}\Big]\Big)$$

$$\nabla_{\mu^{(t)}}\mathcal{L}^{(t+1)} = \sum_i h_i\mu^{(t)}A(v_i^{(t)}) + h_i(1 - \alpha^{(t)}h_i)\mathbb{E}\Big[\theta_i^{(t)}v_i^{(t)}\Big] = 0$$

$$\mu^{(t)*} = -\frac{\sum_i h_i(1 - \alpha^{(t)}h_i)\mathbb{E}\Big[\theta_i^{(t)}v_i^{(t)}\Big]}{\sum_i h_iA(v_i^{(t)})}$$

$$\alpha^{(t)}\sum_i\left(\Big((h_i)^3(A(\theta_i^{(t)}) + (\sigma_i)^2)\Big)\Big(\sum_j h_jA(v_j^{(t)})\Big) - \Big(\sum_j (h_j)^2\mathbb{E}\Big[\theta_j^{(t)}v_j^{(t)}\Big]\Big)(h_i)^2\mathbb{E}\Big[\theta_i^{(t)}v_i^{(t)}\Big]\right) =$$

$$\sum_i\left((h_i)^2A(\theta_i^{(t)})\Big(\sum_j h_jA(v_j^{(t)})\Big) - \Big(\sum_j h_j\mathbb{E}\Big[\theta_j^{(t)}v_j^{(t)}\Big]\Big)(h_i)^2\mathbb{E}\Big[\theta_i^{(t)}v_i^{(t)}\Big]\right)$$

$$\alpha^{(t)*} = \frac{\sum_i\Big((h_i)^2A(\theta_i^{(t)})\Big(\sum_j h_jA(v_j^{(t)})\Big) - \Big(\sum_j h_j\mathbb{E}\Big[\theta_j^{(t)}v_j^{(t)}\Big]\Big)(h_i)^2\mathbb{E}\Big[\theta_i^{(t)}v_i^{(t)}\Big]\Big)}{\sum_i\Big(\Big((h_i)^3(A(\theta_i^{(t)}) + (\sigma_i)^2)\Big)\Big(\sum_j h_jA(v_j^{(t)})\Big) - \Big(\sum_j(h_j)^2\mathbb{E}\Big[\theta_j^{(t)}v_j^{(t)}\Big]\Big)(h_i)^2\mathbb{E}\Big[\theta_i^{(t)}v_i^{(t)}\Big]\Big)}.$$

### A.3 UNIVARIATE OPTIMALITY IN SGD

We now consider a dynamic programming approach to solve the problem. We formalize the optimization problem of $\{\alpha_i\}$ as follows. We first denote $\mathcal{L}_{\min}$ as the minimum expected loss at the last time step $T$ (i.e., under the optimal learning rate).

$$\mathcal{L}_{\min} = \min_{\alpha^{(t)},\alpha^{(t+1)},\ldots,\alpha^{(T-1)}} \mathbb{E}_{\xi^{(t)},\xi^{(t+1)},\ldots,\xi^{(T-1)}}\Big[\mathcal{L}(\theta^{(T)})\Big].$$

Recall that the loss can be expressed in terms of the expectation and variance of $\theta$. Denote $A^{(t)} = (\mathbb{E}\big[\theta^{(t)}\big])^2 + \mathbb{V}\big[\theta^{(t)}\big]$. The final loss can be expressed in terms of $A_{\min}^{(T)}$, obtained by using optimal learning rate schedule:

$$\mathcal{L}_{\min} = \frac{1}{2}hA_{\min}^{(T)} + \sigma^2.$$

As in Theorem 1, we derive the dynamics for SGD *without* momentum:

$$\theta^{(t)} = (1 - \alpha^{(t-1)}h)\theta^{(t-1)} + \alpha^{(t-1)}h\sigma\xi^{(t-1)}$$

$$\Rightarrow \Big(\mathbb{E}\big[\theta^{(t)}\big], \mathbb{V}\big[\theta^{(t)}\big]\Big) = \Big((1 - \alpha^{(t-1)}h)\mathbb{E}\big[\theta^{(t-1)}\big], (1 - \alpha^{(t-1)}h)^2\mathbb{V}\big[\theta^{(t-1)}\big] + (\alpha^{(t-1)}h\sigma)^2\Big).$$

Thus, we can find a recurrence relation of the sequence $A^{(t)}$:

$$A^{(t)} = (1-\alpha^{(t-1)}h)^2\left((\mathbb{E}\big[\theta^{(t-1)}\big])^2 + \mathbb{V}\big[\theta^{(t-1)}\big]^2\right) + (\alpha^{(t-1)}h\sigma)^2 = (1-\alpha^{(t-1)}h)^2A_{\min}^{(T-1)} + (\alpha^{(t-1)}h\sigma)^2.$$

Since $A_{\min}^{(T)}$ is a function of $\alpha^{(T-1)*}$. We can obtain optimal learning rate $\alpha^{(T-1)*}$ by taking the derivative of $\mathcal{L}_{\min}$ w.r.t. $\alpha^{(T-1)}$ and setting it to zero:

$$\frac{d\mathcal{L}_{\min}}{d\alpha^{(T-1)}} = \frac{1}{2}h\frac{dA_{\min}^{(T-1)}}{d\alpha^{(T-2)}} = 0$$

$$\Rightarrow \frac{dA_{\min}^{(T)}}{d\alpha^{(T-1)}} = 0$$

$$\Rightarrow \alpha^{(T-1)*} = \frac{A_{\min}^{(T-1)}}{h(A_{\min}^{(T-1)} + \sigma^2)}.$$

Thus we can write $A_{\min}^{(T)}$ in terms of $A_{\min}^{(T-1)}$ and the optimal $\alpha^{(T-1)*}$:

$$
\begin{aligned}
A_{\min}^{(T)} &= \left( 1 - \frac{A_{\min}^{(T-1)}}{A_{\min}^{(T-1)} + \sigma^2} \right)^2 A_{\min}^{(T-1)} + \left( \frac{A_{\min}^{(T-1)}}{A_{\min}^{(T-1)} + \sigma^2} \sigma \right)^2 \\
&= \left( \frac{\sigma^2}{A_{\min}^{(T-1)} + \sigma^2} \right)^2 A_{\min}^{(T-1)} + \left( \frac{A_{\min}^{(T-1)}}{A_{\min}^{(T-1)} + \sigma^2} \sigma \right)^2 \\
&= \frac{A_{\min}^{(T-1)} \sigma^2}{A_{\min}^{(T-1)} + \sigma^2}.
\end{aligned}
$$

Therefore,

$$
\begin{aligned}
\mathcal{L}_{\min} &= \frac{1}{2} h A_{\min}^{(T)} + \sigma^2 \\
&= \frac{1}{2} h \left( \frac{A_{\min}^{(T-1)} \sigma^2}{A_{\min}^{(T-1)} + \sigma^2} \right) + \sigma^2.
\end{aligned}
$$

We now generalize the above derivation. First rewrite $\mathcal{L}_{\min}$ in terms of $A_{\min}^{T-k}$ and calculate the optimal learning rate at time step $T - k$.

**Theorem 4.** *For all $T \in \mathbb{N}$, and $k \in \mathbb{N}$, $1 \le k \le T$, we have,*

$$
\mathcal{L}_{\min} = \frac{1}{2} h \left( \frac{A_{\min}^{(T-k)} \sigma^2}{k A_{\min}^{(T-k)} + \sigma^2} \right) + \sigma^2. \tag{16}
$$

*Therefore, the optimal learning $\alpha^{(t)}$ at timestep $t$ is given as,*

$$
\alpha^{(t)*} = \frac{A^{(t)}}{h(A^{(t)} + \sigma^2)}. \tag{17}
$$

*Proof.* The form of $\mathcal{L}_{\min}$ can be easily proven by induction on $k$, and use the identity that,

$$
\frac{\left( \frac{ab}{a+b} \right) b}{k \left( \frac{ab}{a+b} \right) + b} = \frac{ab}{(k+1)a + b}.
$$

The optimal learning rate then follows immediately by taking the derivative of $\mathcal{L}_{\min}$ w.r.t. $\alpha^{(T-k-1)}$ and setting it to zero. Note that the subscript $\min$ is omitted from $A^{(t)}$ in Eq.(17) as we assume all $A^{(t)}$ are obtained using optimal $\alpha^*$, and hence minimum. $\qquad\square$

