# OpenReview forum: "Understanding Short-Horizon Bias in Stochastic Meta-Optimization"
_ICLR.cc/2018/Conference — Accept (Poster)_

### Official Review · AnonReviewer1 · 2017-11-21
**Interesting work on meta-optimization**

**Rating:** 7
**Confidence:** 4

**Review:**

The paper discusses the problems of meta optimization with small look-ahead: do small runs bias the results of tuning? The result is yes and the authors show how differently the tuning can be compared to tuning the full run. The Greedy schedules are far inferior to hand-tuned schedules as they focus on optimizing the large eigenvalues while the small eigenvalues can not be "seen" with a small lookahead. The authors show that this effect is caused by the noise in the obective function.

pro:
- Thorough discussion of the issue with theoretical understanding on small benchmark functions as well as theoretical work
- Easy to read and follow

cons:
-Small issues in presentation:
* Figure 2 "optimal learning rate" -> "optimal greedy learning rate", also reference to Theorem 2 for increased clarity.
* The optimized learning rate in 2.3 is not described. This reduces reproducibility.
* Figure 4 misses the red trajectories, also it would be easier to have colors on the same (log?)-scale.
  The text unfortunately does not explain why the loss function looks so vastly different
  with different look-ahead. I would assume from the description that the colors are based
  on the final loss values obtaine dby choosing a fixed pair of decay exponent and effective LR.

Typos and notation:
page 7 last paragraph: "We train the all" -> We train all
notation page 5: i find \nabla_{\theta_i} confusing when \theta_i is a scalar, i would propose \frac{\partial}{\partial \theta_i}
page 2: "But this would come at the expense of long-term optimization process": at this point of the paper it is not clear how or why this should happen. Maybe add a sentence regarding the large/Small eigenvalues?

---

> ### Author Response · Authors · 2018-01-03
> **Thanks for all suggestions. We added the description about optimized schedule. We edited the paragraph in the introduction. All typos corrected.**
>
> Q1: The optimized learning rate in 2.3 is not described. This reduces reproducibility.
> Sorry for such confusion. We use the losses formed by the forward dynamics given in Theorem 1 as training objective, and use Adam to find the learning rate and momentum at each time steps that minimize that training objective. The meta training learning rate is 0.003, and it is trained for 500 meta training steps. We added this description in our revised version.
>
> Q2: Figure 4 misses the red trajectories, also it would be easier to have colors on the same (log?)-scale.
> Meta-descent on 20k is time-consuming to run. Since the visualized hyper-surface is smooth, we expect that meta-descent will behave as expected to converge to the local minimum.  We will add the red trajectory in the next version of the paper.
>
> Q3: Why the loss function looks so vastly different with different look-ahead?
> More number of look-ahead means more optimization steps. The loss will go lower as one trains longer.
>
> Q4. page 2: "But this would come at the expense of long-term optimization process": at this point of the paper it is not clear how or why this should happen. Maybe add a sentence regarding the large/Small eigenvalues?
> Thanks for your suggestion. We modified the entire paragraph as you and reviewer 3 suggested. We believe the current version is clearer.

---

### Official Review · AnonReviewer3 · 2017-11-28
**simplified model demonstrating a problem of meta-learning learning rate**

**Rating:** 6
**Confidence:** 4

**Review:**

This paper proposes a simple problem to demonstrate the short-horizon bias of the learning rate meta-optimization.

- The idealized case of quadratic function the analytical solution offers a good way to understand how T-step look ahead can benefit the meta-algorithm.
- The second part of the paper seems to be a bit disconnected to the quadratic function analysis. It would be helpful to understand if there is gap between gradient based meta-optimization and the best effort(given by the analytical solution)
- Unfortunately, no guideline or solution is offered in the paper.

In summary, the idealized model gives a good demonstration of the problem itself. I think it might be of interest to some audiences in ICLR.

---

> ### Author Response · Authors · 2018-01-03
> **Thanks for your comments! The second part of the paper demonstrates the problem in a more empirical setting. We also offered an potential solution to the problem.**
>
> Q1: The second part of the paper seems to be a bit disconnected to the quadratic function analysis. It would be helpful to understand if there is gap between gradient based meta-optimization and the best effort (given by the analytical solution)
> Ans: The second part of the paper experimentally verified the theory in the first part while generalizing to general neural networks with non-convex problems. It shows that quadratic analysis is a valid model for hyper-parameter optimization. We will work on the flow of the paper with more connection between the two parts of the paper.
>
> Q2: Unfortunately, no guideline or solution is offered in the paper.
> Ans: We agree with the reviewer that in the current version of the paper there’s no solution provided to the problem. We only offered one potential solution to the problem, following theorem 3. Theorem 3 states that the greedy solution is optimal when the curvature is spherical and the noise is codiagonalizable with the curvature. This implies stochastic meta descent could work well with a good enough natural gradient method, such as Kronecker-factored approximate curvature (K-FAC). We will show more experimental results on this subject in a later version.

---

### Official Review · AnonReviewer4 · 2017-12-02
**A clear and well written demonstration of a fundamental issue with meta-optimization.**

**Rating:** 8
**Confidence:** 3

**Review:**

This paper studies the issue of truncated backpropagation for meta-optimization. Backpropagation through an optimization process requires unrolling the optimization, which due to computational and memory constraints, is typically restricted or truncated to a smaller number of unrolled steps than we would like.

This paper highlights this problem as a fundamental issue limiting meta-optimization approaches. The authors perform a number of experiments on a toy problem (stochastic quadratics) which is amenable to some theoretical analysis as well as a small fully connected network trained on MNIST.

(side note: I was assigned this paper quite late in the review process, and have not carefully gone through the derivations--specifically Theorems 1 and 2).

The paper is generally clear and well written.

Major comments
-------------------------
I was a bit confused why 1000 SGD+mom steps pre-training steps were needed. As far as I can tell, pre-training is not typically done in the other meta-optimization literature? The authors suggest this is needed because "the dynamics of training are different at the very start compared to later stages", which is a bit vague. Perhaps the authors can expand upon  this point?

The conclusion suggests that the difference in greedy vs. fully optimized schedule is due to the curvature (poor scaling) of the objective--but Fig 2. and earlier discussion talked about the noise in the objective as introducing the bias (e.g. from earlier in the paper, "The noise in the problem adds uncertainty to the objective, resulting in failures of greedy schedule"). Which is the real issue, noise or curvature? Would running the problem on quadratics with different condition numbers be insightful?

Minor comments
-------------------------
The stochastic gradient equation in Sec 2.2.2 is missing a subscript: "h_i" instead of "h"

It would be nice to include the loss curve for a fixed learning rate and momentum for the noisy quadratic in Figure 2, just to get a sense of how that compares with the greedy and optimized curves.

It looks like there was an upper bound constraint placed on the optimized learning rate in Figure 2--is that correct? I couldn't find a mention of the constraint in the paper. (the optimized learning rate remains at 0.2 for the first ~60 steps)?

Figure 2 (and elsewhere): I would change 'optimal' to 'optimized' to distinguish it from an optimal curve that might result from an analytic derivation. 'Optimized' makes it more clear that the curve was obtained using an optimization process.

Figure 2: can you change the line style or thickness so that we can see both the red and blue curves for the deterministic case? I assume the red curve is hiding beneath the blue one--but it would be good to see this explicitly.

Figure 4 is fantastic--it succinctly and clearly demonstrates the problem of truncated unrolls. I would add a note in the caption to make it clear that the SMD trajectories are the red curves, e.g.: "SMD trajectories (red) during meta-optimization of initial effective ...". I would also change the caption to use "meta-training losses" instead of "training losses" (I believe those numbers are for the meta-loss, correct?). Finally, I would add a colorbar to indicate numerical values for the different grayscale values.

Some recent references that warrant a mention in the text:
- both of these learn optimizers using longer numbers of unrolled steps:
Learning gradient descent: better generalization and longer horizons, Lv et al, ICML 2017
Learned optimizers that scale and generalize, Wichrowska et al, ICML 2017
- another application of unrolled optimization:
Unrolled generative adversarial networks, Metz et al, ICLR 2017

In the text discussing Figure 4 (middle of pg. 8) , "which is obtained by using..." should be "which are obtained by using..."

In the conclusion, "optimal for deterministic objective" should be "deterministic objectives"

---

> ### Comment · AnonReviewer4 · 2017-12-02
> **another comment regarding Figure 1**
>
> I think you could make a figure that much more clearly demonstrates the issue to replace or add to the current Figure 1.
>
> Compute the meta-loss for learning the learning rate for some small problem (e.g. stochastic quadratics). This meta-loss is a 1D function over the learning rate. For a small number of unrolled steps, this function should have minima at low values of the learning rate. You can plot this meta-loss for different numbers of unrolls on the same graph, which should show that the minima of the meta-loss shifts to higher learning rates as you unroll for more steps. This is related to Figure 4, but I think would be a nice way to introduce the problem in an easily digestible picture.

---

> > ### Author Response · Authors · 2018-01-03
> > **We modified the figure 1 as reviewer suggested, along with its descriptions.**
> >
> > We want to thank reviewer again for raising such a great idea to show the problem in a more accessible way. We edited the figure as reviewer suggested.

---

> ### Author Response · Authors · 2018-01-03
> **Thank you very much for all great suggestions. The figures have been edited as reviewer suggested. Citations have been added. Minor typos corrected.**
>
> Q1: Why 1000 SGD+mom steps pre-training steps were needed?
> We want to choose a setting that our observation is less sensitive to which part of training. If we always start looking ahead at zeroth step, then there is a higher chance that the optimal hyperparameter is only fitted to the beginning; whereas if we start at some pre-trained steps, e.g. 1000, then the optimal hyperparameter is more likely to generalize to, say 500 or 5000 steps.
>
> Q2: Which is the real issue, noise or curvature?
> The problem will arise if you have both noise in the objective and different curvature directions. We showed that in a deterministic problem, the greedy optimal learning rate and momentum is optimal as it is essentially doing conjugate gradient, regardless of how many different curvature directions you have. We also showed in theorem 3 that the greedy learning rate is optimal if the curvature is spherical.  On the other hand, if there’s noise in the objective, and there are many different curvature directions the problem will arise. This is because, the noise in the objective forbids one to completely get rid of the loss on a particular direction. Hence, one should always first remove the loss on low curvature directions and then move onto high curvature directions. But short-horizon objective encourages the opposite because high curvature directions gives most rapid decrease in loss. Therefore, both noise in the objective and different curvature directions cause the problem.
>
> Q3: Figure 2: 1. Show fixed learning rate. 2. Thickness of the red curve. 3. Upper bound.
> Figure 2 is edited as reviewer suggests. Also the reviewer is correct that we upper bounded the learning rate to avoid the loss on any curvature direction becoming larger than its initial value, so as to assure the quadratic assumption. We added the description in the revised version.
>
> Q4: Figure 4: 1. Add a color bar to indicate numerical values for the different grayscale values.
> Thanks for the suggestion. We will add it in the next version of our paper.
>
> Q5: Citations:
> We added those citations reviewer mentioned.

---

### Decision · Program_Chairs · 2018-01-29
**ICLR 2018 Conference Acceptance Decision**

**Decision:**

Accept (Poster)

**Comment:**

An interesting analysis of the issue of short-horizon bias in meta-optimization that highlights a real problem in a number of existing setups. I concur with Reviewer 3 that it would be nice to provide a constructive solution to this issue: if something like K-FAC does indeed work well, it would be a great addition to a final version of this paper. Nonetheless, I think the paper would be a interesting addition to ICLR and recommend acceptance.